# Analysis of the Invasion of Acetes into the Water Intake of the Daya Bay Nuclear Power Base

**Xinghao Li [1], Lin Yang [1], Huatang Ren [1,*], Zhaowei Liu [2] and Zeyu Jia [3]**

1. College of Life and Environmental Sciences, Minzu University of China, Beijing 100081, China
2. State Key Laboratory Hydroscience and Engineering, Tsinghua University, Beijing 100084, China
3. YANGTZE Eco-Environment Engineering Research Center, China Three Gorges Corporation, Beijing 100038, China
* Correspondence: renhuatang@muc.edu.cn

**Abstract:** The invasions of marine organisms into the intake of nuclear power plants threaten the normal operation of such plants. Most published numerical models assumed that marine organisms passively follow the current, but such models neglected their biological swimming ability. In this work, adopting a hydrodynamic mathematical model to replicate the flow around the Daya Bay nuclear power base, the invasion characteristics of Acetes were explored by considering the behavior of biological movement. A concept of biological residual current was introduced to describe biological movements that were dominated by both tidal current and biological swimming ability. The biological residual currents near the nuclear power plant were obtained for cases with different nocturnal migration periods (12 h, 13 h, 14 h, 15 h, and 16 h). Using the Lagrangian particle-tracking method, the primary invasion paths of Acetes were obtained, as well as the travel time of Acetes to the intake, based on the biological residual current along monitoring points. The results showed that the invading time for Acetes reaching the water intake of the nuclear power base was significantly decreased when biological migration behavior was considered. When the nocturnal active period was over 13 h, it took only 10 days for Acetes to enter the western waters of Daya Bay from the southwest of Da Lajia Island and then continue migrating to the water intake in the nuclear power base. When the nocturnal active period was less than 13 h, it took more than 20 days for Acetes to travel the same distance. The present work provides a new methodology for the simulation and prediction of the migration of marine organisms.

**Keywords:** Daya Bay; Acetes invasion; EFDC model; biological residual current; Lagrangian particle-tracking method

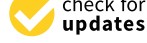



## 1. Introduction

In recent years, as the marine ecosystem has changed, coastal nuclear power stations have frequently experienced incidents of marine organisms and floating debris blocking water intakes. Most coastal nuclear power plant sites choose once-through cooling water systems. When a massive Acetes invasion affects the system, the nuclear power intakes may be clogged, causing a heat sink malfunction [1]. Since the beginning of the 21st century, nuclear power plants in several countries, including the USA, Japan, Israel, and Scotland, have been shut down due to outbreaks of marine organisms [2]. In China, Acetes are among the main organisms that threaten the safety of the cold-water source for nuclear power plants. For example, the Yangjiang nuclear power station [3] and the Daya Bay nuclear power station [4] were both shut down due to outbreaks of Acetes.

Acetes are common marine organisms in oceans that are sensitive to light intensity and water clarity in their habits. Acetes sink to deep water during daytime, with strong light or high water clarity, while they rise up to shallow water at night, with low light or low water clarity [5]. Due to their weak swimming ability, the migration of Acetes is

predominated by passive motivation that follows the water flow. Specifically, during the daytime, the gradient of light in a vertical direction causes the Acetes to migrate toward the bottom and cling there until they float up at night. These biological habits make the migration of Acetes different from those of dissolved inorganic matter. It is important to advance the understanding of the Acetes invasion at the intake of a power station to ensure the safe operation of power stations.

An aquatic organism's migration velocity is the summation of the velocity of the surrounding water and its velocity relative to the surrounding water, i.e., a comprehensive effect of hydrodynamic and biological movement characteristics. In the investigation of hydrodynamic properties, early works focused on small water bodies with near-stationary flow velocities, such as lakes, using a zero-dimensional model. That model only considered the mass changes relating to local sources/sinks of mass [6]. In 1987, Saint-Venant derived a set of partial differential equations to represent non-constant flow, which became the foundation of one-dimensional hydrodynamic models. Currently, the one-dimensional models are adopted for simple water bodies, such as inland river networks and water bodies flowing vertically along a river channel. Wang et al. [7] proposed a one-dimensional model to replicate the Yunxi River network in Hangzhou, China, which predicted the water quality of the complex river network in the Qiantang River diversion project.

To overcome the limitations of one-dimensional models, researchers developed two-dimensional hydrodynamic models in the horizontal plane or the vertical plane. Lopes et al. [8] adopted the MIKE21 model to replicate Lake Ria de Aveiro and obtained the planar distribution of the concentration of dissolved oxygen. Martin et al. [9] developed the vertical two-dimensional (2D) model CE-QUAL-W2 to represent De Gray Lake in the USA. That model effectively predicted the evolution of vertical water temperature distribution and the hydrodynamic properties of the lake. The vertical 2D model or the horizontal 2D model ignore the differences in another dimension. Therefore, the horizontal 2D model is only valid for wide and shallow waters, e.g., shallow lakes. Similarly, the vertical 2D model ignores the lateral difference and is only valid for narrow and deep waters, e.g., river-type reservoirs.

For more complex water areas, the models should account for the variations along all three spatial dimensions to obtain accurate results. Researchers have developed a series of 3D models, such as the Finite-Volume Coastal Ocean Model (FVCOM) [10], the Princeton Ocean Model (POM) [11], and the Environmental Fluid Dynamics Code (EFDC) model [12]. Alarcon et al. [13] adopted the EFDC model to construct a 3D hydrodynamic model for Biscayne Bay and Coral Cable Canal in Florida, USA. They explored the effects of mean synoptic and extreme sea level rises on coastal inundation in the Coral Gulch basin. The 3D models are valid for various waters, as they consider a wider range of hydrodynamic properties.

With respect to the oceans, the movements of mass particles or organisms relative to their surrounding water are more difficult than hydrodynamic properties to simulate. To replicate non-soluble substances with different densities from the water, researchers introduced the settling velocity of substances to investigate the process of settling or resuspending, by analogy with sediment settling. Based on the COHERENS model, Chen et al. [14] developed a three-dimensional Bohai-Yellow-East Sea summer circulation model to determine the possible sources of red tide in the Yangtze River estuary and its adjacent waters. That model simplified algae as hydrodynamically swimming particles and considered the upward migration of algae. Based on the numerical investigation, they found that the source of red tide in the Yangtze River estuary was most likely from the Taiwan Strait and northeastern Taiwan Island, which lacked field data to verify these conclusions. The settling velocity model considers the net buoyancy effect of the material, which results from physical mechanisms and is assumed as a constant. However, in most cases, the biological property cannot be simply modeled by settling velocity [15].

To obtain a further understanding of the growth, mutation, reproduction, and drift of marine organisms in practical situations, researchers have replicated biological pathways

using particle track techniques. Based on the Gulf of Mexico (GOM) circulation model, Johnson et al. [16] showed that jellyfish abundance and distribution were influenced by seasonal variations in the inner circulation. Using a model that coupled particle tracking with a hydrodynamic model, North et al. [17] considered the effect of the vertical swimming behavior of oyster larvae on their dispersal distribution in the Chesapeake Bay, and found that the modeled biological behavior had a larger impact on the spatial distribution of transport than did the interannual differences in circulation patterns. Using the fine-resolution Antarctic model (FRAM) model to predict shrimp transport in the ocean, Hofmann et al. [18] explored the transport of Antarctic krill, representing krill by Lagrangian particles to investigate their traces under surface complex flow fields. They discovered that the krill population in the western Antarctic Peninsula is the source for the krill population around South Georgia Island. Using the Harvard Ocean Prediction System (HOPS) to simulate the circulation processes in and around the Scotia Sea, Fach et al. [19] verified that most of the Antarctic krill in the South Georgia Sea east of the Scotia Sea were the original source of the krill individuals around the west side of the Antarctic Peninsula. Meanwhile, the drift simulations in the HOPS showed that the location and timing of krill particle releasing significantly affected the possibility of krill transporting from further north in the Drake Passage to South Georgia. In conclusion, the seasonal variability of the complex flow field and the biological habits of shrimp are critical factors in improving the accuracy of numerically simulated shrimp migration in the ocean.

Although numerical models that consider biological habits have been applied in oceans, the relative velocity to their surrounding water adopted a constant value in existing research. In reality, the relative velocity is in a dynamic change process, along with biological habits. Therefore, the constant relative velocity may result in large errors in the simulation of the absolute velocity field or the migration period.

The Daya Bay nuclear power base is located in the northern part of the South China Sea, which includes the Ling'ao nuclear power station and the Daya Bay nuclear power station. The area of Daya Bay is nearly 1000 square kilometers. The ecological environment of Daya Bay is suitable for Acetes [20]. The high growth rate, high reproductive capacity, and poor autonomous swimming ability of Acetes are responsible for outbreaks of Acetes bloom. When a massive Acetes invasion affects nuclear power intakes, they may be clogged, causing a heat sink malfunction [2,21] and resulting in risks to operation. From 2015 to 2016, there were two incidents of Acetes invasion at the intakes in the Daya Bay nuclear power base, which led to the shutdown of the nuclear power plant units [22].

In this paper, EFDC models were used to study the dynamics of Acetes clogging the Daya Bay nuclear power station. The present work analyzed the migration characteristics of Acetes in the waters near the nuclear power plant, taking into account the diurnal biological habits of the Acetes and the influence of the nuclear power plant's abstraction and discharge. The migration paths, migration times, and invasion potential of Acetes were simulated for various situations. During the modeling process, the concept of biological residual current was newly proposed by considering the diurnal habits of the shrimp. This study provides an early warning method to prevent the invasion of Acetes in the intakes of nuclear power plants.

## 2. Methods

### 2.1. Numerical Model

Integrating several mathematical models, Hamrick [12] developed the environmental fluid dynamics computer code (EFDC) model. Since 1992 when EFDC model was released, it has been continuously upgraded and improved. It has become one of the most widely used mathematical models in the field of water environment research. In recent decades, the EFDC model has been widely adopted for the simulation of flow fields in the Taihu lakes [23], the Changtan reservoirs [24], the Jinhae bays [25], and other waters in China. It has also been implemented to investigate various factors affecting the flow field, such as

the wind field [26], temperature [27], and power plant abstraction and discharge [28]. The governing equations of the EFDC model are stated as follows.

Momentum conservation equations:

$$\frac{\partial}{\partial t}mHu + \frac{\partial}{\partial x}(m_y Huu) + \frac{\partial}{\partial y}(m_x Huv) + \frac{\partial}{\partial \sigma}(mwu) - \left(mf + v\frac{\partial m_y}{\partial x} - u\frac{\partial m_x}{\partial y}\right)Hv$$
$$= -m_y H\frac{\partial}{\partial x}(g\zeta + p) - m_y\left(\frac{\partial h}{\partial x} - \sigma\frac{\partial H}{\partial x}\right)\frac{\partial p}{\partial \sigma} + \frac{\partial}{\partial \sigma}\left(mH^{-1}A_v\frac{\partial u}{\partial \sigma}\right) + Q_u \tag{1}$$

$$\frac{\partial}{\partial t}mHv + \frac{\partial}{\partial x}(m_y Huv) + \frac{\partial}{\partial y}(m_x Hvv) + \frac{\partial}{\partial \sigma}(mwv) + \left(mf + v\frac{\partial m_y}{\partial x} - u\frac{\partial m_x}{\partial y}\right)Hu$$
$$= -m_x H\frac{\partial}{\partial y}(g\zeta + p) - m_x\left(\frac{\partial h}{\partial y} - \sigma\frac{\partial H}{\partial y}\right)\frac{\partial p}{\partial \sigma} + \frac{\partial}{\partial \sigma}\left(mH^{-1}A_v\frac{\partial v}{\partial \sigma}\right) + Q_v \tag{2}$$

Mass conservation equation:

$$\frac{\partial}{\partial t}(m\zeta) + \frac{\partial}{\partial x}(m_y Hu) + \frac{\partial}{\partial y}(m_x Hv) + \frac{\partial}{\partial \sigma}(mw) = 0 \tag{3}$$

where $t$ denotes time; $u$, $v$, $w$ are the velocity components along the x-direction, y-direction, and vertical direction in the orthogonal curve coordinates, respectively; $m_x$ and $m_y$ are the coordinate transformation factors, $m = m_x m_y$; $H = h + \zeta$ ($H$ is the instantaneous total water depth, $\zeta$ is the water surface elevation and $h$ is the distance between average sea level and the sea floor); $A_v$ is the vertical turbulent eddy viscosity coefficient; $p$ is the relative hydrostatic pressure; $f$ is the Coriolis force coefficient; and $Q_u$ and $Q_v$ are the power source-sink terms.

To accurately represent the undulating seafloor topography, the EFDC model adopts the $\sigma$ coordinates in the vertical direction, as follows:

$$\sigma = \frac{z^* + h}{\zeta + h} = \frac{z^* + h}{H} \tag{4}$$

where $z^*$ is the vertical coordinate from vertical reference horizontal datum and $\zeta$ is the water surface elevation above the vertical reference datum.

*2.2. Computational Domain, Boundary Conditions and Cases*

Figure 1 depicts the computational domain in the present study, which is located at 114.5° E—114.9° E, 22.3° N—22.5° N, including the entire Daya Bay. Rectangular grids of 300 m × 300 m were adopted to discretize the horizontal computational domain. The total number of grids is 7227. In the vertical direction, a standard $\sigma$ coordinate system was adopted, and the water column was divided equally into 10 layers from bottom to top. Shore boundaries and bathymetry were obtained from 1:35,000 chart data provided by the relevant management authorities in Daya Bay. Bathymetry at each grid point was obtained from interpolation. Daya Bay is about 20 km wide from east to west and 30 km long from north to south. The total area of the sea is about 650 square kilometers, with a coastline of 92 km. The water depth in Daya Bay deepens gradually from north to south, with an average depth of 11 m throughout the area and reaching a depth of 20 m at the mouth of the bay.

Offshore boundary conditions were determined by the harmonic analysis of nine principal tidal components (M2, K1, O1, S2, P1, SA, N2, Q1, and K2) [20]. Observations showed that most of Acetes blockages around the nuclear power station occurred in November. Thirteen months, from 1 January 2020 to 1 February 2021, were simulated in the present work. For the nuclear power plant, either the intake or the outlet was generalized as a cell. The water flux was set as 300 m³/s both at the intake and the outlet. The effects of inland river confluence on the study area were neglected because most of the rivers are located north of computational domain and were far away from the outlet.

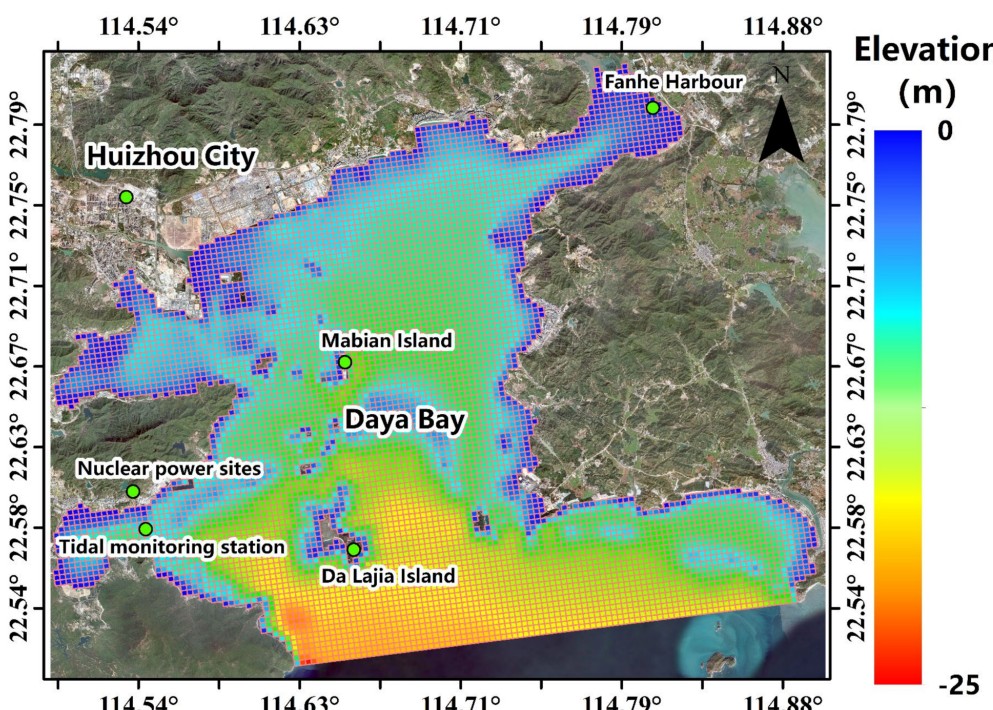

**Figure 1.** Computational domain and monitoring stations.

The bottom roughness height, $z_0$, was set as 0.02 m and the time step was 6.9 s. Tides are related to the lunar calendar, as tide is mainly attributed to the moon's circulation movement. All calendars referred to lunar calendar. Under normal weather conditions, the nighttime duration in Daya Bay is approximately 13 h during November. Five cases with different nighttime durations (12 h, 13 h, 14 h, 15 h, and 16 h) were investigated, considering that cloudy and rainy weather may decrease light intensity.

### 2.3. Acetes Migration

For organisms with a strong ability to swim, the rate of biological horizontal migration must be considered, as their migrations are significantly affected by predations and other biological activities. It is difficult to quantify the movement of these organisms, as their migration are predominated by subjective movement. Conversely, Acetes and jellyfish have weak autonomous swimming ability and their horizontal migrations are predominated by the passive movement of the water flow. Hence, their horizontal migration velocities are negligible compared to the water flow velocity [29]. In studies of marine migration in deeper waters, diurnal vertical migration of animals can reach over 100 m, whereas in the small water depths (<20 m) of the study area, the vertical migration process of organisms can be negligible. There was a lack of quantitative information on the vertical migration of Acetes in this region, so the study used a simplified model that did not consider the vertical migration process of Acetes, i.e., it was assumed that Acetes migrated to the seafloor during the day and rose to the surface waters at night and migrated horizontally with the current.

### 2.4. Biological Residual Current

The transportation of organisms in a water body is affected not only by the flow of the water body but also by their migration habits in the water body. Organisms' migration is expected to be influenced by their surrounding environments, e.g., water temperature and light, as well as by their survival needs, including predation and escape. In the analysis of organism transportation in water bodies, the concept of biological residual current was proposed by integrating biological habits and water flow. Accounting for biological habits, the biological residual current is the average rate of biological migration in a single biological movement cycle.

The method to quantify the biological residual current of Acetes could be developed from the water column residual flow equation. The migration velocity vector of Acetes for one biological activity cycle (24 h) is summed up for the biological residual current $U_M$ as follows:

$$\overrightarrow{U_M} = \frac{1}{T} \int_{t_1}^{t_2} \overrightarrow{u_{acete}}\, dt \tag{5}$$

In the equation, $T$ is the time for one biological movement cycle; $t_1$ and $t_2$ are the beginning and ending times of biological transport respectively; $x$ and $y$ are the coordinates; $\overrightarrow{u_{acete}}$ is the component of flow rate in the x direction (east–west) and y direction (north–south) at time $t$; the daytime migration velocity of Acetes is $\overrightarrow{u_{acete}} = 0$ and the nighttime migration velocity is equal to the water velocity, i.e., $\overrightarrow{u_{acete}} = (u,v)$.

### 2.5. Lagrange Particle-Tracking Method

To obtain the invasion migration path and time, the Lagrangian analysis method models the Acetes as particles to track the movement of particles in the fluid field. The Lagrangian particle-tracking method in EFDC Explorer 8.4 software could predict the distribution of particles and the migration of invasion organisms. As the particle-simulation method aimed to investigate the invasion threat of the Acetes, it was generally assumed that Acetes were migrating long distances from the outer sea and that the easiest or most dangerous infiltration points needed to be identified at the open boundary. According to this objective, Figure 2 depicted multiple release sites at the western inlet from south to north in the sequence of a1, a2, a3, a4, a5, and a6. According to a parametric study, the results indicated that releasing particles at position a2 resulted in the shortest time to reach the intake. This observation was consistent with the analysis of the biological residual current, in which point a2 was located in the biological residual current channel. This consistency indicated that the present model could accurately replicate the physical model. The particle release date was set as the beginning of the inlet cycle for all cases, and the moment of release was aligned with the biological residual current channel flow condition to ensure the most vulnerable conditions for the Acetes to invade the intake.

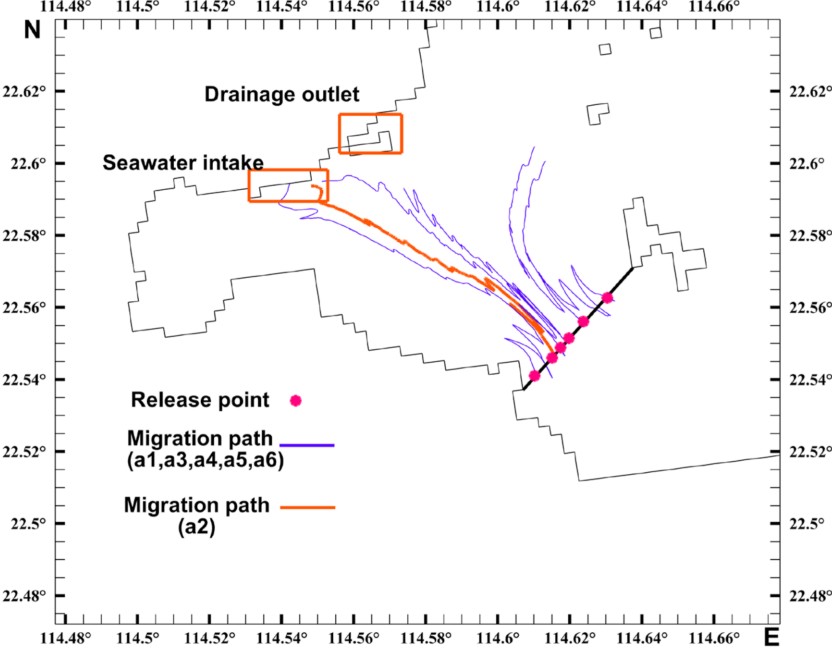

**Figure 2.** Lagrangian particle-release locations and migration paths (14 h daytime cases).

*2.6. Validation of Hydrodynamic Property*

2.6.1. Hydrodynamic Property

At the Daya Bay tidal monitoring station, operated by the State Oceanic Administration of China, the water levels were recorded from November 2020 to January 2021 and compared to the water levels predicted by using the EFDC model (Figure 3). The comparisons illustrated that the maximum error between the prediction and measured tide levels was less than 10%. The phases and amplitudes of predicted and the measured water levels were in good agreement, which indicated the high reliability and validity of the present model.

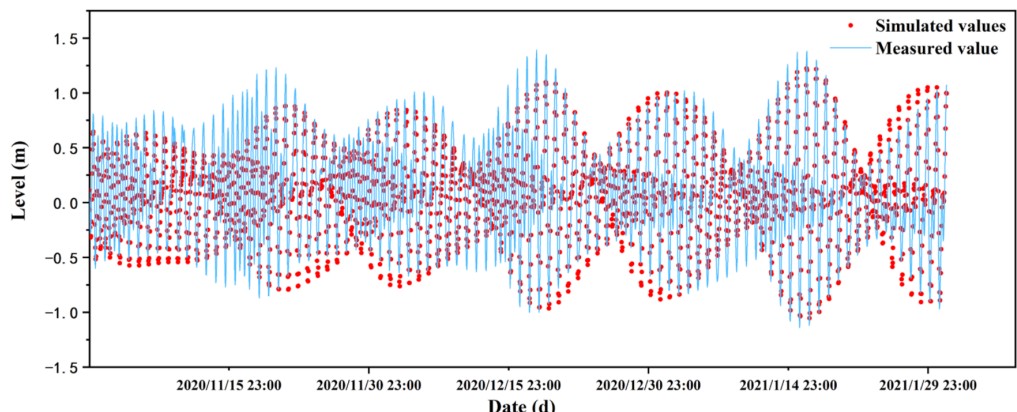

**Figure 3.** Comparison of simulated and measured water levels at the Daya Bay tidal station.

2.6.2. Acetes Invasion

Using the Lagrangian particle-tracking method, the results for the biological residual current along the monitoring points showed that the fastest time of invasion for the Acetes was approximately 7 to 10 days. In 2015 and 2016, there were two serious blockage accidents at the Daya Bay nuclear power site. Table 1 shows the recording dates for these two accidents and the losses due to these accidents.

**Table 1.** The timing and consequences of the Acetes invasion.

| Date | Description of the Event |
|---|---|
| 10 January 2015 | In this accident, works cleaned approximate 1.3 tonnes of Acetes in the drum net backwash drain at the nuclear power site's Ling'ao 2 unit. Units 1 and 2 had to operate with lower power. |
| 9 January 2016 | In the circulating water filtration system, the number of Acetes detected was much higher than the tolerance level of safety design. Units 1 and 2 had to be shut down. |

The date of clogging by Acetes on 10 January 2015 was actually 20 November 2014 in the lunar calendar. Based on the results for the cases with 13 h of night-migration time, the Acetes invasion happened approximately on the last day of the outflow period, as the Acetes reached the intake within one cycle.

The Acetes clogging accident at the Daya Bay nuclear power base on 8 January 2016 occurred on 28 November in the lunar calendar, which was the eighth day of the inflow period. The weather conditions in the Daya Bay waters in January 2016 were continuously cloudy and rainy from 3 January to 6 January. Such weather increased the duration for the night migration of Acetes, providing conditions for them to reach the attraction range at the intake in one inflow period. This observation was consistent with the results predicted with 14 h for the night migration.

The consistency of the prediction and the actual observation in the physical model illustrated the reliability of the numerical model developed in the present work. The numerical model provided a theoretical approach for the prediction of Acetes clogging at the Daya Bay nuclear power station.

## 3. Results and Discussion

### 3.1. Flow Field Characteristics

Marking the flow directions with arrows, Figure 4 depicts the flow fields of the rising tides in Daya Bay. The rising tide flowed from south to north and the falling tide flowed in the opposite direction. These were the main characteristics of a typical reciprocating flow, which were consistent with the results from previous studies [30]. During the rising period, the water flowed from outside Daya Bay into the bay mouth (A), which could be divided into two parts. The first part entered the western waters of Daya Bay through the western side (B) of Da Lajia Island, which further split into two flows near the Daya Bay nuclear power base, i.e., one toward the Da peng'ao Bay mouth (C) and the other heading northward, mainly into Huizhou Port (D). The other part of the tide headed northward from the eastern side of Da Lajia Island and merged with part of the western side of the current; then, it flowed into Fanhe Harbor (E) in the northeastern part of Daya Bay.

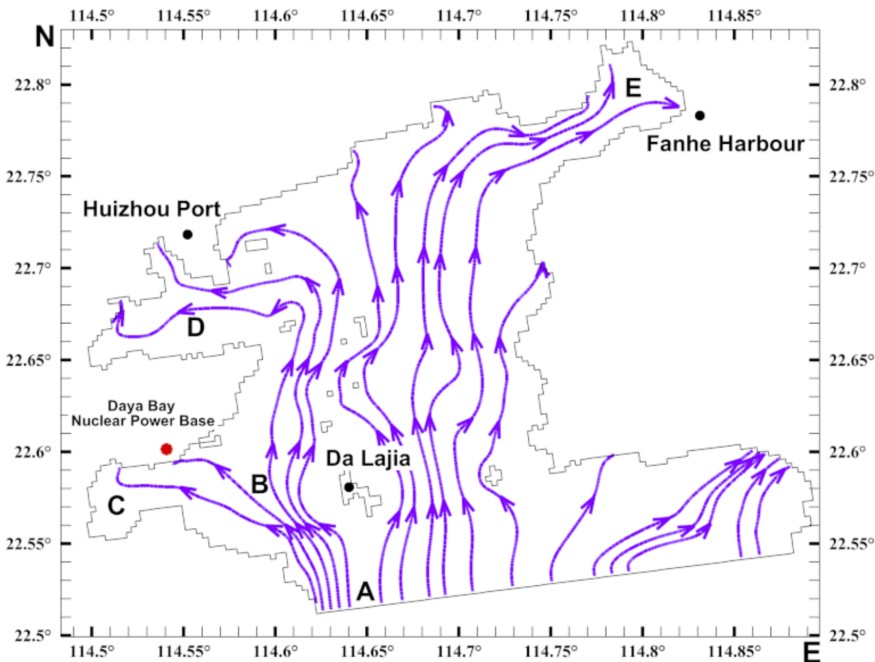

**Figure 4.** Schematic diagram of inflow direction.

In low tide, the currents flowed out of the computational domain in the opposite direction of the rising tide. The overall flow velocity in Daya Bay was slow, mostly distributed in the range of approximately 0.1 to 0.2 m/s. The flow velocities near Da Lajia Island and the Daya Bay mouth varied between 0.4 m/s and 0.5 m/s. In the Dapeng'ao area, where the Daya Bay nuclear power base is located, the flow velocity was slow and less than 0.1 m/s in some local areas. Acetes have a weak ability to swim on their own and highly depend on the water currents for their migration within the bay. The weak outflow condition in Daya Bay was beneficial for the aggregation of Acetes populations and contributed to Acetes outbreaks. Meanwhile, the weak hydrodynamic condition was a disadvantage for the rapid migration of Acetes. Hence, the waters adjacent to the water intake were the main areas for Acetes clogging.

A comparison of the surface flow field and the bottom flow field in Figure 5 shows that the surface flow velocity in Daya Bay was greater than the bottom flow velocity during the rise and fall moments, but the difference in flow velocities was smaller, indicating that the difference in the vertical direction of the flow field in the Daya Bay region was small. As shown in Figure 6, the difference in flow velocity from layer 8 to layer 10 was less than 2%. The subsequent analyses of the Daya Bay flow field and the residual currents field

were conducted for the surface layer of Daya Bay, as the migration of Acetes occurred in the upper layers.

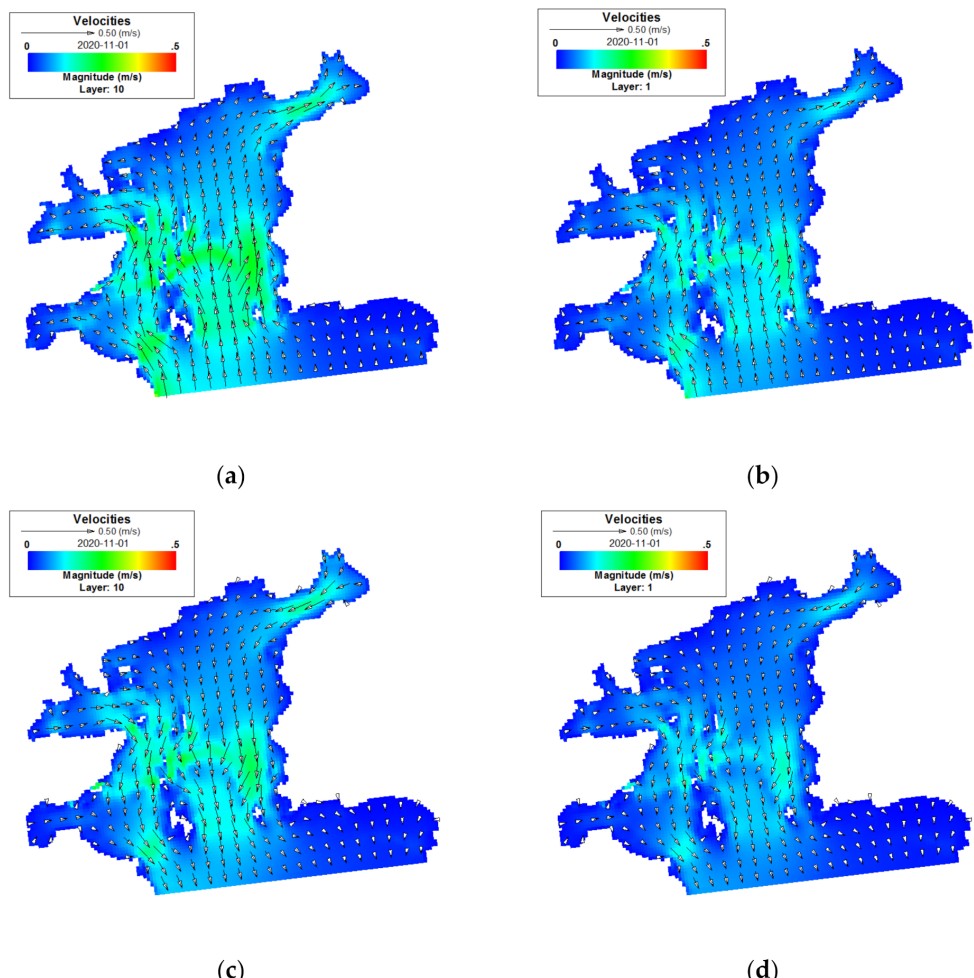

(**a**)                                                                    (**b**)

(**c**)                                                                    (**d**)

**Figure 5.** Tidal current field in Daya Bay: velocity distributions of (**a**) surface flow and (**b**) bottom flow during fastest high tide; velocity distribution of (**c**) surface flow and (**d**) bottom flow during fastest low tide.

### 3.2. Characteristics of the Biological Residual Current of Acetes

In the area near the intake of the nuclear power plant, the predicted residual currents for different cases indicated that the pattern of residual currents repeated several times in November. Each repetition was defined as one period. In each period, the half period as residual currents approached the intake was defined as the inflow period, and the other half period as the residual currents flowed far away from the intake was defined as the outflow period. Each half cycle lasted for a few nights.

At the moment when the residual flow velocity reached its peak during the inlet cycle, the distribution of the residual flow velocity for five cases was determined. The maximum residual velocity occurred at a similar moment in the five cases, as presented in Figure 7, in which the differences between the maximum residual velocities were less than 1 m/s. The inflow period was around 10 days and the outflow period was 6 days in all cases. This result may be due to a combination of astronomical tides and tidal waves in the marginal seas, which were influenced by topography and water depth [31].

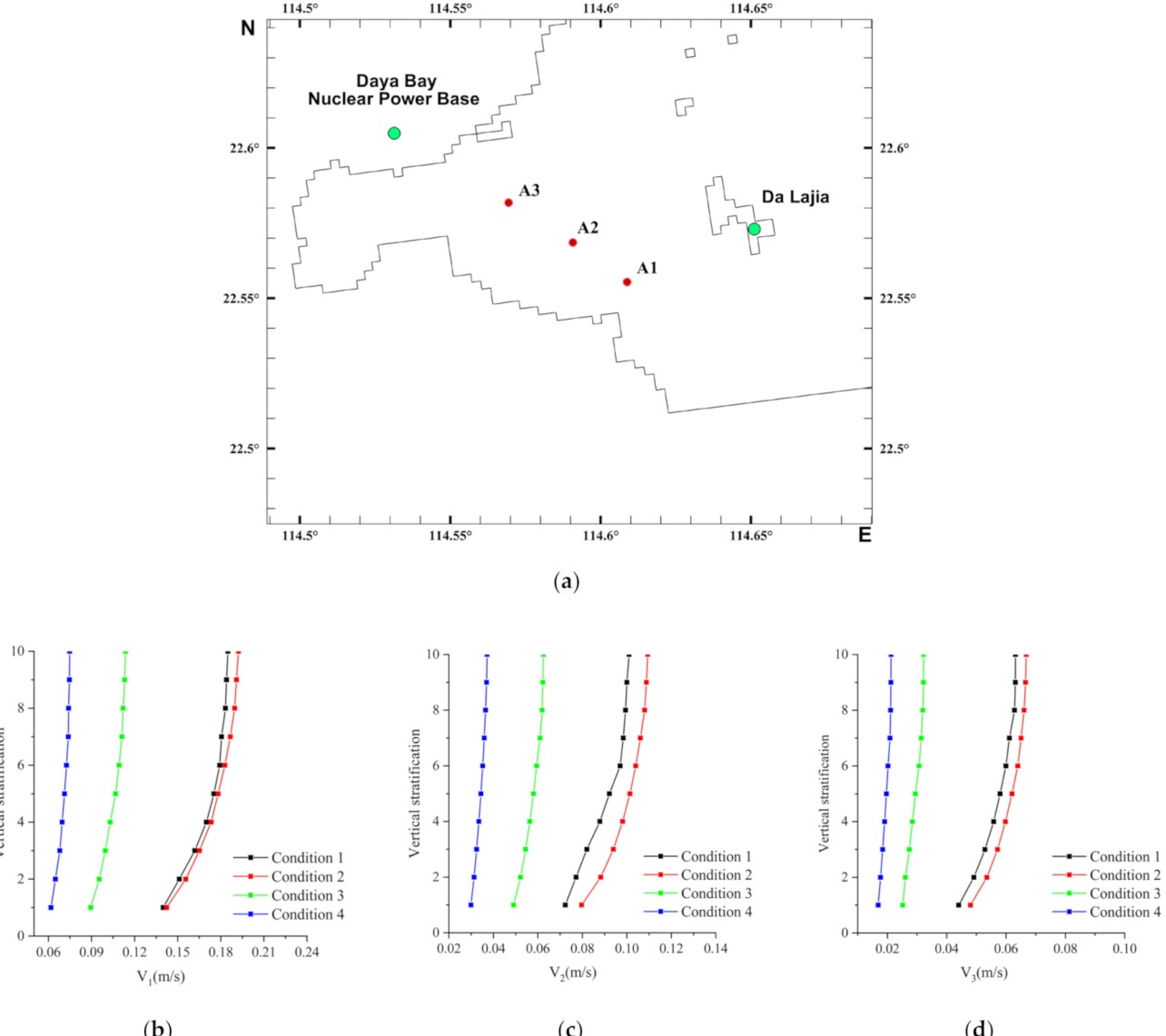

**Figure 6.** Velocity distribution in vertical direction at each characteristic point: (**a**) typical location to show velocity; (**b**) magnitude of vertical stratified flow velocity at point A1; (**c**) velocity at point A2; (**d**) velocity at point A3. (Condition 1: the moment when the rising tide flows fastest; Condition 2: the moment when the falling tide flows fastest; Condition 3: the moment when the tide is at its lowest and the water level is stable; Condition 4: the moment when the tide is at its highest and the water level is stable).

### 3.3. Analysis of the Migration Pathways of Invasive Acetes

Figure 8 shows that there is a migration channel for Acetes during the inflow period from the western part of the inlet of Daya Bay to the water intake. The predicted residual flow velocity was about 2–3 cm/s at night in the southwestern area of Da Lajia Island, which was about 1 cm/s at the middle part of the channel. The residual flow velocity was about 2–3 cm/s in the area closer to the intake and reached a maximum value of 3.5 cm/s at the intake of the nuclear power plant. The presence of circulation on both sides of the residual flow channel was not conducive to the migration of particles. Therefore, one could conclude that the Acetes passed through the biological residual current channel near the intake during the inflow period, which led to an invasive blockage by the Acetes. Based on the simulation results of all cases, Figure 8f indicates that there was a fan-shaped area around the water intake of the nuclear power plant, in which the residual flow was always

toward the intake in both the inflow and the outflow periods. The center of this fan-shaped area located at the water intake of the nuclear power plant had a radius of 2–3 km. This observation indicated that, when the Acetes migrated within the area, they quickly reached the intake, due to the pumping effect of the fluid.

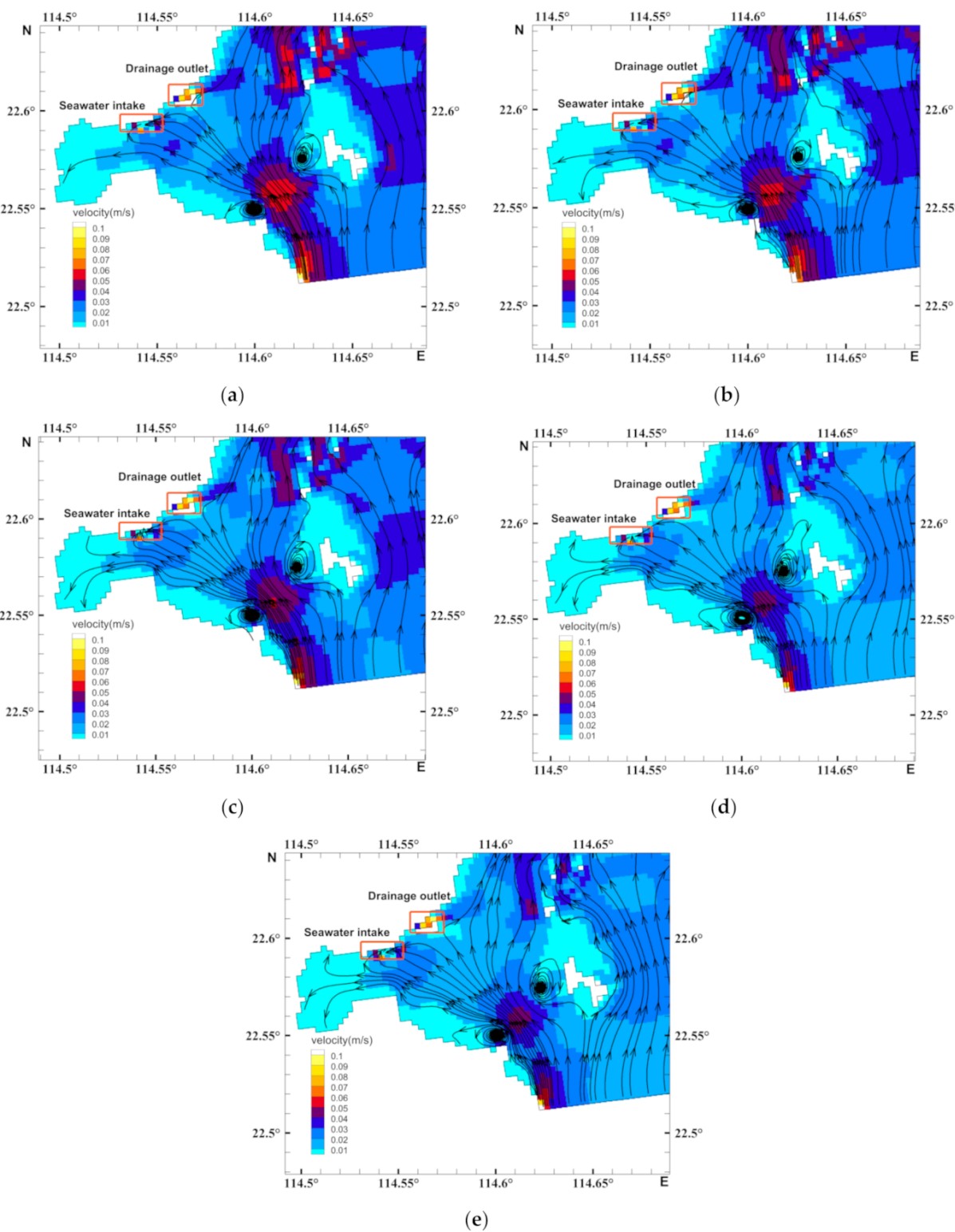

**Figure 7.** Maximal residual velocity field near water intake for various cases: (**a**) 12 h nighttime; (**b**) 13 h nighttime; (**c**) 14 h nighttime; (**d**) 15 h nighttime; and (**e**) 16 h nighttime.

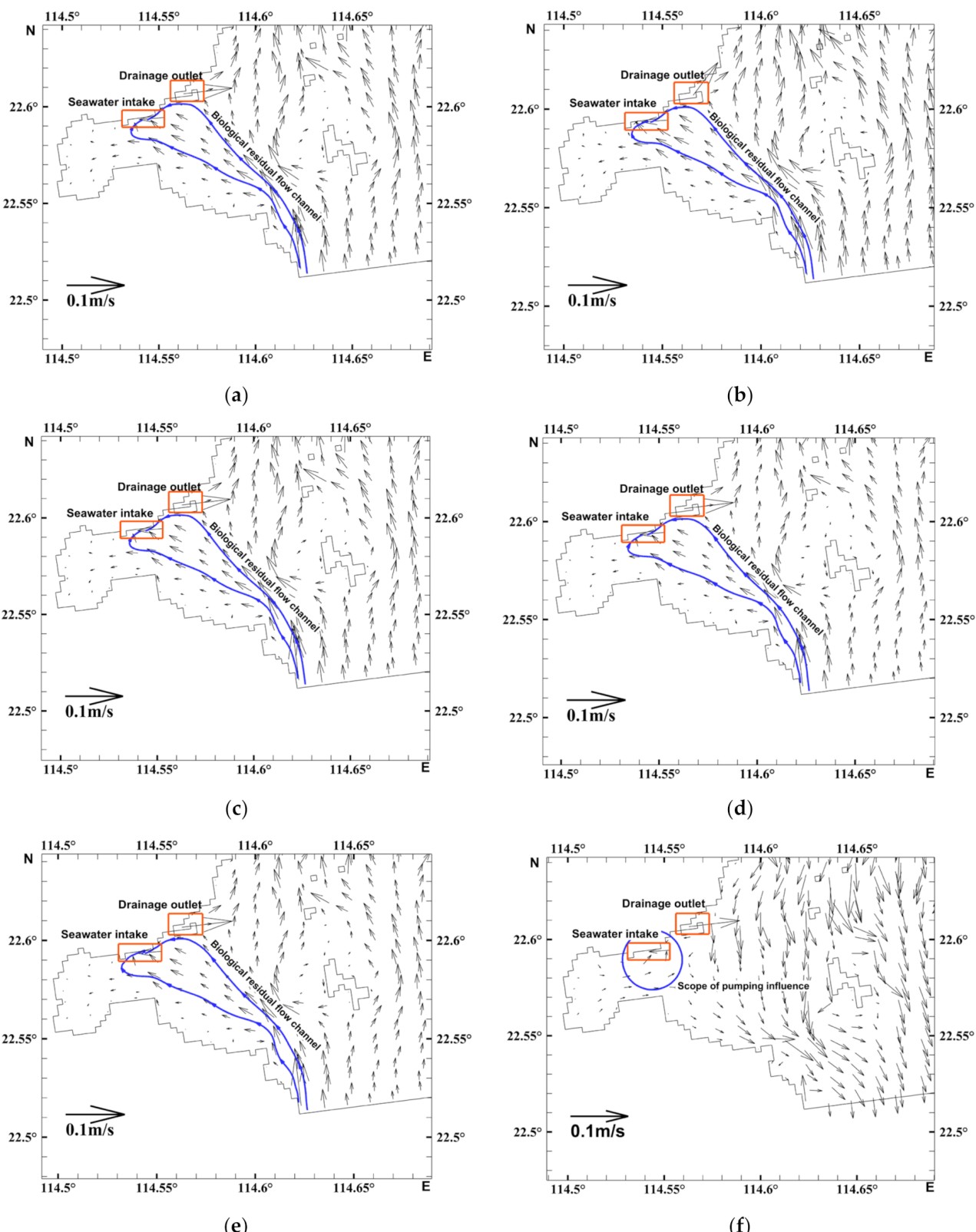

**Figure 8.** Residual flow field of inflow periods: (**a**) 12 h night-migration period; (**b**) 13 h night-migration period; (**c**) 14 h night-migration period; (**d**) 15 h night-migration period; (**e**) 16 h night-migration period; (**f**) residual flow field of outflow period with a 14 h night-migration period.

### 3.4. Timing Analysis of the Acetes Invasion

Figure 9 displays the biological residual current channel representing the Acetes invasion path, which was marked as red curve L. Figure 8 shows the monitoring points that were arranged along the path from south to north in the sequence of L0, L1, L2, L3, L4, L5, L6, L7, L8, L9, L10, and L11. The distances between monitoring points L0 to L8 were 1 km for each segment. In the area close to the intake, the variation in residual flow velocity had a large value, as the intervals between monitoring points L8 to L11 were set at 500 m each. The averages of the biological residual current rates at the monitoring points were calculated to investigate the variation in the biological residual current rate over time along the pathway. The results are depicted in Table 2.

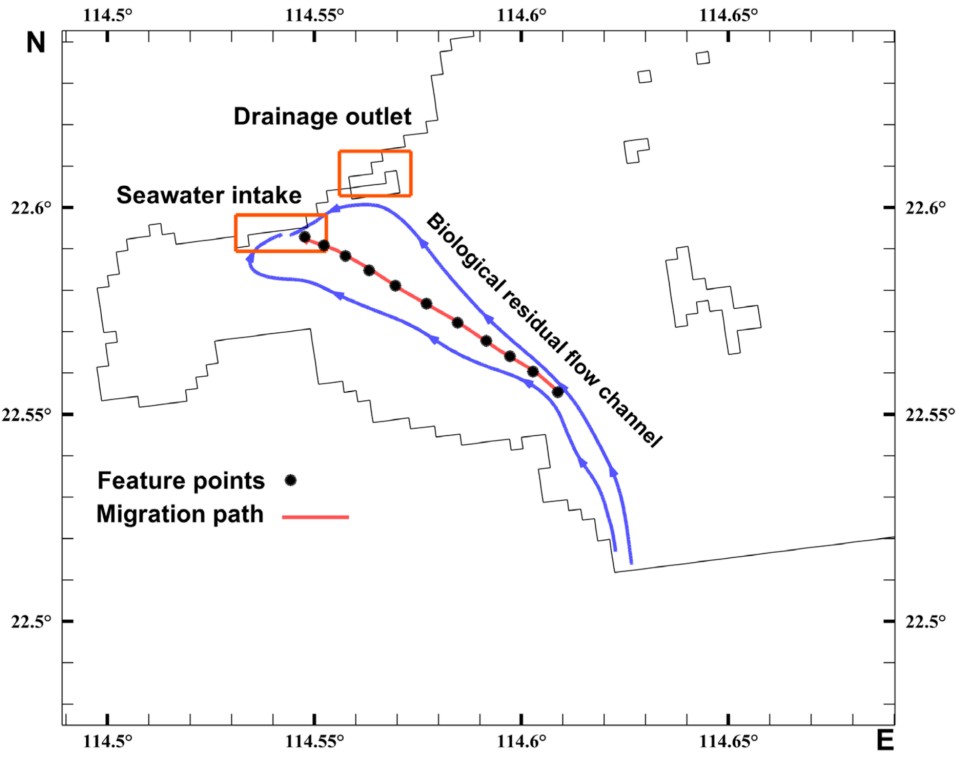

**Figure 9.** Migration path and monitor points arrangement.

**Table 2.** Variations of inflow periods for various cases (dates are recorded in the lunar calendar).

|  | 12 h | 13 h | 14 h | 15 h | 16 h |
|---|---|---|---|---|---|
| **Calculation time/h** | 18:00–6:00 (The following day) | 18:00–7:00 (The following day) | 17:00–7:00 (The following day) | 16:00–7:00 (The following day) | 16:00–8:00 (The following day) |
| **Date of entry into the stream** | 4–14 November | 5–14 November | 5–15 November | 5–14 November | 5–14 November |
| **Duration of inflow days/d** | 10 | 9 | 10 | 9 | 9 |
| **Date of departure** | 11.14–11.20 | 11.14–11.20 | 11.15–11.20 | 11.14–11.20 | 11.14–11.20 |
| **Duration of outflow days/d** | 6 | 6 | 5 | 6 | 6 |
| **Date of peak residual flow rate** | 11.09 | 11.10 | 11.09 | 11.09 | 11.10 |

According to the predicted velocity fields of the biological residual current, there should be a significant variation in the residual velocity values along the biological residual current channel, as shown in Table 3. At both ends of sections of the migration path, the velocities of the biological residual current were higher than those at the middle section. The biological residual current rate was higher when the seawater passed through the western inlet from outside Daya Bay. The biological residual current gradually decreased as the water area opened up and the flow diverted. Then, due to the pumping effect at the intake, the biological residual current increased as it approached the intake. Compared with the numerical results for cases with different night-migration durations, there was less variation in the magnitude of the average biological residual current velocity at the

monitoring points far away from the intake. However, the variations of the residual flow velocity were higher at the characteristic points close to the intake.

**Table 3.** Average biological residual current velocities at monitored points.

| | | L0 | L1 | L2 | L3 | L4 | L5 | L6 | L7 | L8 | L9 | L10 | L11 |
|---|---|---|---|---|---|---|---|---|---|---|---|---|---|
| Average biological residual current rate/cm·s$^{-1}$ | 12 h | 2.89 | 2.80 | 2.46 | 1.86 | 1.40 | 1.10 | 0.95 | 1.09 | 1.46 | 1.64 | 1.90 | 3.44 |
| | 13 h | 2.92 | 2.74 | 2.31 | 1.76 | 1.35 | 1.08 | 0.93 | 1.05 | 1.39 | 1.58 | 1.97 | 3.50 |
| | 14 h | 3.51 | 3.39 | 2.83 | 2.15 | 1.66 | 1.33 | 1.13 | 1.28 | 1.68 | 1.75 | 2.47 | 3.53 |
| | 15 h | 3.57 | 3.35 | 2.72 | 2.09 | 1.63 | 1.31 | 1.12 | 1.26 | 1.62 | 1.70 | 1.95 | 3.50 |
| | 16 h | 3.54 | 3.28 | 2.62 | 2.02 | 1.59 | 1.29 | 1.10 | 1.24 | 1.58 | 1.67 | 1.93 | 3.49 |

The migration paths were divided into three subsections: A, B, and C. Subsection A, between points L0 and L4, had an average residual flow velocity of 2.5 cm/s. Subsection B, between points L4 and L8, had an average residual flow velocity of 1.30 cm/s. Due to the suction effect at the intake, the biological residual current had a larger valve as it approached the intake, with an average value of 2.19 cm/s for subsection C between points L8 and L11. One could conclude that subsection C was the main area affected by water intake. The result from Jiang et al. [32] demonstrated that the sudden outbreak and aggregation of Acetes populations were the primary reason for the blockage incidents. Thus, in the simulation of Acetes migration, the dispersed Acetes in the investigated area were simplified as particles. Their migration times were equivalent to the times for particles to reach the water intake. To calculate the migration times for the dispersed Acetes, the average migration velocity of the prawn through the paths (LA, LB, LC) were obtained for different night-migration times. Since the residual flow in subsection A was always toward the intake and the migration time was about two days, the average migration velocities were used to determine whether the prawn could pass through subsections B and C in the present inflow period. If not, the dispersed Acetes followed the outflow fluid and became far away from the intake, moving toward the intake in the next inflow period. Table 4 presents the migration of Acetes in different night-migration periods.

**Table 4.** Migration times of Acetes in the different night-migration periods.

| Nocturnal Migration Hours | Average Migration Speed through the Segments/km·d$^{-1}$ | | | Duration of the Inflow Cycle/d | Determining Whether an Incoming Flow Period Can Reach Section C(Y/N) | Through Various Periods/d | | | Time of Arrival at the Water Intake/d |
|---|---|---|---|---|---|---|---|---|---|
| | A | B | C | | | A | B | C | |
| 12 h | 0.91 | 0.52 | 0.99 | 10 | N | 2.6 | >15 | 4.1 | >20 |
| 13 h | 0.99 | 0.54 | 1.03 | 9 | N | 2.4 | >15 | 3.0 | >20 |
| 14 h | 1.18 | 0.71 | 1.37 | 10 | Y | 1.9 | 5.8 | 3.0 | 10.7 |
| 15 h | 1.18 | 0.75 | 1.44 | 9 | Y | 2.0 | 5.5 | 2.9 | 10.4 |
| 16 h | 1.24 | 0.78 | 1.50 | 9 | Y | 1.9 | 5.3 | 2.8 | 9.9 |

Table 4 shows that the Acetes could not reach subsection C within one inlet period for cases with 12 h and 13 h of night-migration times. In cases with 12 h and 13 h of night-migration times, the biological residual current rates had shorter inflow periods and smaller values than in the other cases. As a result, the cases with 12 h and 13 h of night-migration times required more than 15 days for the dispersed Acetes to pass through subsection B, with a low risk of invasion by Acetes. When the migration time at night was longer than 14 h, the Acetes could pass through subsections B and C in one inflow period and, thus, enter subsection A. In addition, the migration time extension increased the average migration speed. The minimum time was about 10 d for the Acetes migrating from the western inlet to the intake of the nuclear power base.

*3.5. Lagrangian Particle Analysis*

After determining the initial release location of the prawn particles, the prawn invasion times were obtained for various cases. To evaluate the effect of the diurnal nature of the prawn on the invasion migration, a special case with a migration time of 24 h was simulated for comparison. Figure 10 depicts the intrusion paths obtained from the simulation with

different migration durations (13 h, 14 h, 15 h, 16 h, and 24 h). Figure 10a shows that the particles moved reciprocally with the tide in the western part of Daya Bay in cases with night-migration times of 12 h, 13 h, and 24 h, and the reciprocal feature was more obvious than it was in other cases with different night-migration times. As 12 h and 13 h were close to the ebb and flow cycle of the Daya Bay tides, the particles migrated northwestward while the tide flowed toward the intake during the ebb cycle, and migrated southeastward while the tide flowed away from the intake during the outflow cycle. In cases with 12 h and 13 h of nighttime migration, the reciprocal paths of the Acetes invasion demonstrated the effects of the tides on particle migration. For cases with 14 h, 15 h and 16 h of nighttime migration, Acetes particles could reach the water intake. This also showed that the invasion route gradually shifted northward for cases with longer nighttime migration times.

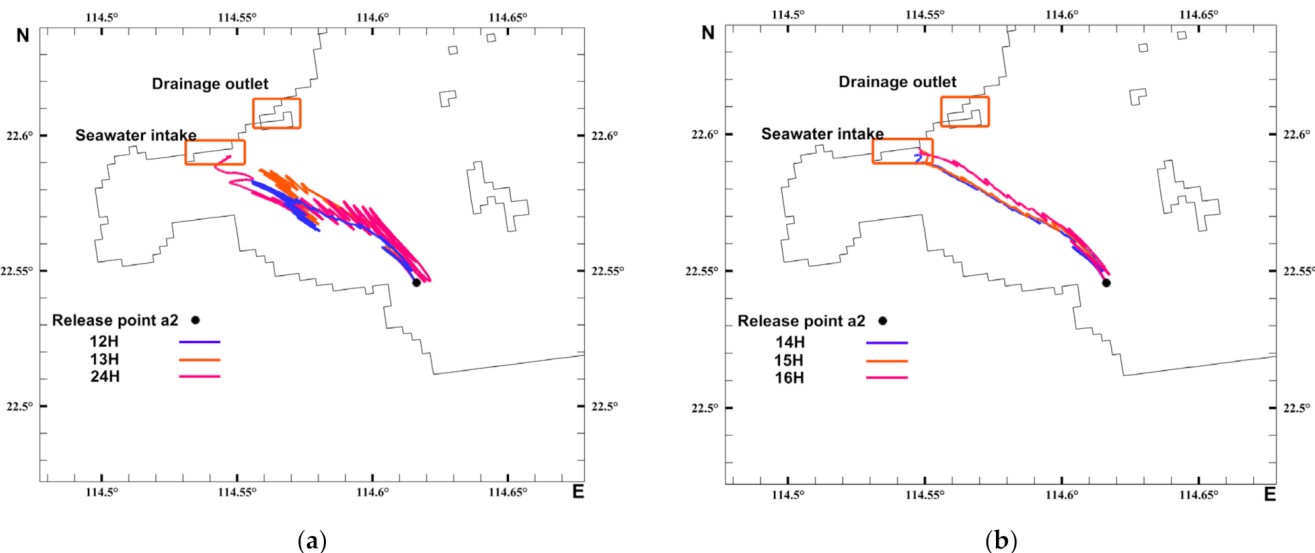

(a)  (b)

**Figure 10.** Invasion paths of Acetes with different night migration time: (**a**) 12 h, 13 h, 24 h; (**b**) 14 h, 15 h, and 16 h of night-migration times.

For cases with 12 h and 13 h of night migration, the statistical results in Table 5 indicated that the Acetes particles could not reach the intake within one residual flow cycle. These observations were consistent with the results obtained from the analysis of the biological residual current. These observations indicated that the threat of Acetes invasion was low with a night-migration time of less than 13 h. For situations of 14 h, 15 h, and 16 h night migration, Acetes could invade from the western inlet to the intake of the nuclear power station within 10 days. The invasion times with 14 h, 15 h, and 16 h of night migration were much less than the invasion time with 24 h of night migration. This observation suggested that the nocturnal migration characteristics of the Acetes increased the speed and likelihood of invasion. The shortest time required to invade the intake was 6.5 days, with 16 h of night-migration time.

**Table 5.** Invasion times of Acetes for cases with various night-migration times.

|  | 12 h | 13 h | 14 h | 15 h | 16 h | 24 h |
|---|---|---|---|---|---|---|
| **Acetes invasion time/d** | >15 | >15 | 8 | 7 | 6.5 | 12 |

## 4. Conclusions

In this paper, a hydrodynamic model of Daya Bay was constructed, based on the EFDC model, to explore the flow field in the sea adjacent to the Daya Bay nuclear power base. Considering the diurnal vertical migration characteristics of Acetes, the biological residual current characteristics under various night-migration durations were investigated. Adopting the Lagrangian particle-tracking method, the migration time for Acetes from

the western inlet to the intake was analyzed for various night-migration durations. The following conclusions were drawn:

(1) The overall hydrodynamics of Daya Bay were weak, with all flow velocities below 0.5 m/s. As Acetes migrated with the current, the small residual flow velocity of the water column prevented the rapid migration of Acetes and made it easy for them to form aggregations of populations. The comparative analysis found that the intake and discharge affected the flow field in a range of approximately 2 to 3 km around the intake and discharge.

(2) Based on the diurnal vertical migration characteristics of the Acetes, the concept of biological residual current was introduced. This concept assumed that the Acetes only migrated along with the water at night. Thus, different nocturnal migration durations (approximately 12 to 16 h) were established for the analysis of the biological residual current of Acetes in the sea near the nuclear power base. During the inlet cycle, a biological residual current channel was formed from the inlet to the intake of the nuclear power plant in the southwest of Da Lajia Island. The residual current velocity was around 1 cm/s in the middle of the channel, and increased to 2–3 cm/s in the area approaching the intake, and reached 3.5 cm/s at the intake of the nuclear power plant.

(3) According to the analysis of the biological residual current paths, the potential invasion paths of Acetes were identified to estimate the migrate time from the western inlet to the intake under various nocturnal migration durations. The results showed that the diurnal vertical migration characteristics of the Acetes increased their invasion potential, especially with nocturnal migration durations over 13 h. The Acetes could migrate from the inlet to the intake within 10 days.

(4) To investigate the migration traces of the Acetes, the Lagrangian particle-tracking method was adopted to obtain the invasion path and the invasion time. When the night-migration time was less than 13 h in winter, the Acetes could not reach the water intake in one cycle. The daily migration distance of the particles gradually increased as the night-migration time increased. As a result, the invasion path gradually shifted to the north, leading to a shorter invasion time, within 6.5 d in cases with 16 h of night migration. The threat of invasion was related to the duration of nighttime migration. The location of the invasion and the overall calculation indicated that winter rainy weather was the worst situation for the Acetes' invasion to the water intake.

This study improved the mathematic model of an aquatic organism's migration. Based on biological habit, we adopted different migration velocities in a biological activity cycle. As a new concept, biological residual current was introduced to simulate the long-time movement of an organism, which simplifies simulation in coastal areas with periodic characteristics.

There is still potential for further refinement of the study, due to the limitations of the observational data. As there are many factors influencing the invasion of Acetes, future research may investigate other factors that influence the invasion of Acetes, such as (1) meteorological factors, including wind direction, weather extremes, and daily changes in light hours; (2) the growth and breeding cycle of Acetes (the premise of the nuclear power plant blockage requires an outbreak of Acetes populations, but the growth and breeding pattern of Acetes in the region remains to be explored); and (3) the impact of nuclear power plant warm-water drainage (the warm-water drainage from the nuclear power plant has an impact on water temperature changes in the Daya Bay region, and the impact of water temperature changes on the growth and migration of Acetes in the region has not yet been reported). In addition, the model's boundary conditions and the information required for parameter calibration can be supplemented by additional environmental monitoring in the Daya Bay area, which will further improve the accuracy of the simulations.

**Author Contributions:** Conceptualization, X.L. and L.Y.; methodology, X.L. and L.Y.; software, X.L. and L.Y.; validation, X.L. and L.Y.; formal analysis, X.L., L.Y. and H.R.; investigation, H.R., Z.L. and Z.J.; resources, H.R., Z.L. and Z.J.; data curation, X.L. and L.Y.; writing—original draft preparation, X.L. and L.Y.; writing—review and editing, H.R. and Z.L.; visualization, X.L.; supervision, H.R., Z.L. and Z.J.; project administration, H.R., Z.L. and Z.J.; funding acquisition, Z.L. and Z.J. All authors have read and agreed to the published version of the manuscript.

**Funding:** This research was funded by the National Natural Science Foundation of China [NSFC Grants 51979142]; by financial support from the China Three Gorges Corporation (WWKY-2020-0031); and by the Research Program of the State Key Laboratory of Hydroscience and Engineering [2020-KY-01].

**Data Availability Statement:** Not applicable.

**Conflicts of Interest:** The authors declare no conflict of interest.

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
