# Peer review of "Analysis of the Invasion of Acetes into the Water Intake of the Daya Bay Nuclear Power Base"

_water, doi:10.3390/w14223741_

Round 1

Reviewer 1 Report

I believe this work is attractive to the potential readers but need to be significantly improved. I give some comments on this paper and hope they will help the authors to improve their paper. Please see the details in the document I have uploaded.

Author Response

Dear reviewer:

Thanks for your decision and constructive comments on my manuscript. We have carefully considered the suggestion of Reviewer. We have tried our best to improve and made some changes in the manuscript.

Please check the attachments we have uploaded.

Reviewer 2 Report

This paper analyzed Acetes invasion at the water intake of Daya bay nuclear power. To describe the Acetes velocity, the concept of biological residual current is introduced based on the diurnal vertical migration characteristics of the Acetes. The results show that the biological residual current of the Acetes increases its invasion potential especially for scenarios with nocturnal migration durations over 13h. To investigate the Acetes migration traces, the Lagrangian particle tracking method, considering Acetes’ habit, is adopted to obtain the invasion path and the invasion time. The daily migration distance of the particles gradually increases as the night migration time increase. The paper contributes biological residual current for aquatic animals’ transportation, which reflects the influence of biological habit.

There are some small problems, which should be solved before it is considered for publication. If the following problems are well-addressed, this reviewer believes that the essential contribution of this paper is important for Eco-hydrological modeling.

(1) In Figures 3,4,6,7,8, and 9, the latitude and longitude need to mark the unit i.e. degree symbol. In figure 7, the unit of the velocity vector also need to mark the unit in legend.

(2) In the paper, night-time migration velocity is equal to the water surface velocity. It needs to show the vertical velocity difference is very small.

(3) The simulation didn’t consider the influence of water temperate difference. The authors need to explain the reason.

(4) In Figure 7(b), “scope of influence” is suggested to be revised as “scope of pumping effect”, according to its interpretation.

(5) The pumping of intake of the nuclear power plant has significant impact on the flow field and Acetes migrate within the range of 2-3 kilometers nearby. Add more information about the boundary conditions setting of the water intake.

(6) Some figures are not clear enough. The accuracy of those figures should be further improved. It is suggested to supplement the topographic elevation map of the bay.

(7) Supplement the computing grid information, including quantity and size.

Author Response

(The authors gave the same response as above.)

Round 2

Reviewer 1 Report

The authours have addressed all my concerns, and I have no further comment for this paper.  I just find the figure 3 and figure 6 can not be clearly saw at present.

Author Response

Dear Editors and Reviewers:

Thanks for your decision and constructive comments on my manuscript. We have improved the clarity of the figures and uploaded the original figures to meet the requirements. Thank you very much for your continued support of this article.

Thank you and best regards.

Yours sincerely,

Huatang Ren